# Liver Regeneration and Immunity: A Tale to Tell

**DOI:** 10.3390/ijms24021176

**Published:** 2023-01-07

**Authors:** Nicola Di-Iacovo, Stefania Pieroni, Danilo Piobbico, Marilena Castelli, Damiano Scopetti, Simona Ferracchiato, Maria Agnese Della-Fazia, Giuseppe Servillo

**Affiliations:** 1Department of Medicine and Surgery, University of Perugia, Piazzale L. Severi 1, 06129 Perugia, Italy; 2Centro Universitario di Ricerca sulla Genomica Funzionale (C.U.R.Ge.F.), University of Perugia, 06123 Perugia, Italy

**Keywords:** liver regeneration, hepatocytes proliferation, hepatocytes growth, IL-6, HGF, IL-17, Th17/Treg, MoMϕ

## Abstract

The physiological importance of the liver is demonstrated by its unique and essential ability to regenerate following extensive injuries affecting its function. By regenerating, the liver reacts to hepatic damage and thus enables homeostasis to be restored. The aim of this review is to add new findings that integrate the regenerative pathway to the current knowledge. An optimal regeneration is achieved through the integration of two main pathways: IL-6/JAK/STAT3, which promotes hepatocyte proliferation, and PI3K/PDK1/Akt, which in turn enhances cell growth. Proliferation and cell growth are events that must be balanced during the three phases of the regenerative process: initiation, proliferation and termination. Achieving the correct liver/body weight ratio is ensured by several pathways as extracellular matrix signalling, apoptosis through caspase-3 activation, and molecules including transforming growth factor-beta, and cyclic adenosine monophosphate. The actors involved in the regenerative process are numerous and many of them are also pivotal players in both the immune and non-immune inflammatory process, that is observed in the early stages of hepatic regeneration. Balance of Th17/Treg is important in liver inflammatory process outcomes. Knowledge of liver regeneration will allow a more detailed characterisation of the molecular mechanisms that are crucial in the interplay between proliferation and inflammation.

## 1. Introduction

The liver is one of most important organs responsible for monitoring homeostasis in the mammalian body. It is involved in physiological processes such as: bile production, plasma protein synthesis, nutrient absorption and detoxification, immunity and vitamin storage [1,2,3,4,5]. Moreover, it is fundamental in metabolic regulation of carbohydrate, protein, and lipid homeostasis [6]. The liver is also the central immunological organ, able to activate the immune system in response to the circulating antigens [7]. Not surprisingly, the cellular immune component in the liver is very large, both lymphoid (e.g., Natural Killer) and myeloid (e.g., Kupffer cells) [8,9]. In this regard, it is clear that the structural and functional integrity of the liver must always be guaranteed. The ratio of liver to body weight must be constant over time and it is known to be approximately 2% [10]. Many authors assume that to keep a proper liver size there are very peculiar molecules, such as bile acids, which act as sensor of the hepatostat control system [11,12]. Therefore, it is clear how the various insults that liver suffers -mechanical, metabolic or biological- can lead to the loss of homeostasis and can undermine health.

The maintenance of homeostasis is guaranteed by a perfect coordination of the immune system. Several immune cells in the liver detect antigens and, as sensors, trigger the immune response. Among the parenchymal cells of the liver there are hepatocytes and cholangiocytes, while the hepatic stellate cells (HSCs) and liver sinusoidal endothelial cells (LSECs) are the non-parenchymal counterpart [13]. Nevertheless, a continued exposure to microbial antigens from the gastrointestinal tract exposes the liver to the risk of an over-activated immune response. Evidence of hepatic tolerance was published as early as 1969 [14]. The liver has the task of ensuring on the one hand the activation of the immune system and on the other the triggering of tolerance mechanisms. The perfect balance of these processes guarantees physiological homeostasis. Indeed, it has to be considered the central role played by liver in glucose and lipids metabolism.Mass loss in liver regeneration is accompanied by hypoglycaemia and transient fat accumulation that indicates a possible role to promote and control liver regeneration [15,16].

This review analyses the main mechanism that is triggered in the liver during severe stress: liver regeneration (LR). The main objective of this review will be to introduce alternative molecular pathways that complement those already known in the field of liver regeneration. Precise coordination of these processes enables the liver to respond optimally to the stimuli it perceives.

## 2. Liver Injuries

When the liver is severely compromised, homeostasis is lost. It could be affected by distinct injuries that determine acute and chronic liver diseases.

### 2.1. Acute Liver Disease

Following rapid and unexpected hepatic damage, the acute liver injury can occur if the liver fails to react. Acute damage is characterised by an increase in infiltrating and activated neutrophils, oxidative stress and massive liver cell death [17,18].

An excessive hepatocyte loss results in tissue degeneration leading to acute liver failure (ALF). There are many different causes of ALF, and the most common cause is drug-induced liver injuries (DILI) [19]. An adverse reaction during liver metabolic function in biotransformation of xenobiotics is the main cause of DILI [20]. In developed countries, the main cause of ALF is paracetamol overdose—in approximately 55% of cases—while it is almost completely absent in Asia and Africa [21,22]. In addition, the causes of acute liver failure include viral hepatitis (e.g., HAV, HBV, HEV, HBV co-infection with HDV, CMV, Epstein-Barr virus, HIV), which can lead to long-term damage, and hepatic ischemia, which may occur followed by reperfusion after liver resection surgery [23,24,25]. Fortunately, most frequently, the reparative process that is triggered after the initiation of liver injuries is resolutive [26].

### 2.2. Chronic Liver Diseases

When a damage affects the liver for more than six months, it is defined as chronic liver diseases (CLDs). Clinically, CLDs undergo a process of liver damage and regeneration, which leads to progressive fibrosis that can degenerate into cirrhosis [27,28]. The outbreak and progression of chronic liver disease includes several different causes, such as virus infections, environmental and genetic factors [29,30]. The main causes of CLDs include: hepatitis B (HBV) and C (HCV) viruses, alcohol consumption and obesity [31,32]. From an immunological point of view, chronic liver diseases are characterised by a constant activation of infiltrating cells and hepatic stellate cells (HSCs), which induce liver damage and fibrous tissue insertions.

Fibrosis is dynamic process. During the connective tissue accumulation, the extracellular matrix is continuously degraded and remodelled in a finely regulated process that may culminate in cirrhosis or in the restoration of normal liver organisation and function. Even when cirrhosis has also become visible, remodelling can allow partial recovery [33,34]. The onset of cirrhosis is one of the most common etiological factors for hepatocellular carcinoma [35]. In order to preserve its integrity, the liver responds to injury by attempting to resolve it through the liver regeneration process [36].

## 3. Liver Regeneration

Approximately 80% of the liver mass consists of hepatocytes, which guarantee its perfect functioning. The hepatocytes are normally in a quiescent state (G0 phase), but they re-enter cell cycle following various types of perturbation [37]. Partial hepatectomy (PH) is the most common approach to study liver regeneration, as an in vivo experimental model of proliferation in rats, mice and also zebrafish [38,39,40].

Liver regeneration is the re-establishment of structural or physiological integrity, which was previously lost. In the framework of the liver, it is actually a compensating enlargement by the residual tissue. After PH, the liver is restored by the residual hepatocytes, without activation of a progenitor cell [41]. Nevertheless, bone marrow cells can generate approximately 20% of endothelial and non-parenchymal cells during liver mass restoring [42]. Liver regeneration occurs through the interaction of different molecular pathways. Several factors affect the final success. Following PH, the amount of excised liver determines the activation of different proliferative pathways. When the resection affects 1/3 of the liver, the hepatocytes quickly react by expanding their volume, initiating hypertrophy. Instead, if the resection affects 2/3 of the mass, the hepatocytes enter the S phase, proliferate and induce hyperplasia [43]. A third mechanism is triggered if the PH affects 80–90% of the liver and it has been well characterized in both mouse and zebrafish. Hepatocytes in this case are not enough to restore the liver, so biliary epithelial cells are activated and converted into functioning hepatocytes [44,45,46]. 

## 4. Liver Regeneration Step by Step

The regenerative process can be illustrated mainly in three phases: initiation, proliferation and termination (Figure 1). Liver regeneration has been extensively studied over the years. The players involved in it are really several and the interactions between the different pathways involved during the three phases need further investigation. In the following section, we will explore new insights and alternative pathways complementing the classic ones.

### 4.1. Initiation Phase

The initiation phase involves molecules characteristic of innate immunity. In a normal liver, the hepatocytes are in G0, the resting phase of the cell cycle. It is important to have a trigger to enter the cycle (G1 phase) and replicate. The response achieved after PH has been described to be similar to the wound healing repair process, with slight modifications [47]. No conclusive evidence is available to support the initiation of the response in the LR. In this stage, part of the activation of the cascade is certainly related to early hemodynamic alterations in the quantity and quality of portal vein flow. Consequently, the portal volume increases, generating shear stress. To compensate, the hepatic arterial buffer response (HABR) is implemented, which, through vasodilation, reduces arterial flow [48,49]. An additional potential candidate could be lipopolysaccharide (LPS), which is commonly released by the degradation of intestinal bacteria [50]. LPS as part of the molecular patterns associated with pathogens (PAMPs) is recognized by Toll-like receptors (TLRs), but there is no evidence that TLRs are involved in the initiation phase, while it is dependent on the activation of the shared intracellular cascade. Specifically, a knock-out (KO) mouse model for one or several TLRs do not result in altered priming, while the lack of MyD88—an adapter for intracellular signalling of TLRs—results in impaired LR [51,52]. Following the priming, Kupffer cells significantly produce two key cytokines at this stage: tumour necrosis factor alpha (TNF-α) and interleukin 6 (IL-6). TNF-α and IL-6 production is mediated by the activation of the nuclear factor kappa-light-chain-enhancer of activated B cells (NF-κB) [50]. In addition, hepatocytes express the TNF receptor 1 (TNFR1) and are able to self-activate in autocrine or paracrine mode [53]. NF-κB binds the promoter of IL-6, controlling its production and secretion [54]. Hepatocytes interacting with Kupffer cells, become responsive to the proliferative stimuli and contribute to the regenerative process progression [55]. Without the priming phase, the following reaction to growth factors would not be achievable [56]. Notch and beta-catenin translocation to the nucleus that already occurs 5–15 min after PH, or the activation of the transcription factors NF-κB and Activator Protein 1 (AP-1) which takes place within 30 min were observed during the initiation phase [57] (Figure 1).

### 4.2. Proliferation Phase

The increase of IL-6 occurring at the beginning of liver regeneration determines the activation of proliferation, acting on Janus kinase (JAK) and inducing phosphorylation on the signal transducer and activator of transcription 3 (STAT3) [58]. Hepatocytes enter in the G1 phase, and the signalling mediated by the growth factors ensures their progression in the cell cycle. The main actors in this scenario are the hepatocyte growth factor (HGF) and the specific ligands of the epidermal growth factor receptor (EGFR). Mitogenic signalling activated by these factors induces significant growth in hepatocytes.

In addition to the mentioned factors, other important signalling molecules contribute to the growth pathway. These are accessory molecules, which act as modulators of EGFR signalling and include: fibroblast growth factor (FGF), vascular endothelial growth factor (VEGF), platelet derived growth factor (PDGF), insulin, bile acids and norepinephrine [59]. The transforming growth factor alpha (TGF-α), is also involved in liver regeneration stimulating DNA synthesis in hepatocytes [60]. Through the c-met receptor, HGF-dependent signals are activated, while the activation of the other players of the LR depend on EGFR [61,62]. In this way, EGF pathway is not restricted by a single ligand, but the other replaces the lack of one [63]. In summary, the tyrosine kinases receptors EGFR and c-met are the two main pathways to activate the signal cascade that leads to the LR process. Although the two pathways do not show common pathways, they compensate mutually for the lack of the other one by implementing a successful LR. When the signal cascade is simultaneously interrupted in both pathways, the result is hepatic failure [64] (Figure 1).

### 4.3. Termination Phase

The termination phase takes place when the original weight of the liver is reached, which results in the homeostasis restoration. The hepatostat allows a perfect integration of processes that leads to the ending of LR and the recovery of liver function. The terminating phase is still largely unexplained. The sequence of control events that restrains proliferation and triggered growth is still being explored.

The liver/body weight ratio is a fundamental parameter for the correct and complete liver mass restoring [65]. In this respect, it has been seen that transplanting from a donor animal with a smaller liver to the receiver, entails a re-organisation that allows the transplanted liver to reach the weight of the recipient [66]. Several sources suggest that the proper functioning of the hepatostat is provided by mammalian target of rapamycin (mTOR), demonstrating that, post PH, the rapamycin induced inhibition of mTOR is able to block DNA replication [67,68]. The mTOR pathway controls protein translation, and thus regulates cell growth by integrating different components such as hormones, growth factors, metabolic requirements, energy demand and nutrients [69]. As early as the end of the 1980s, the transforming growth factor-beta (TGF-β) was assumed to play a role in complete regenerative process. Specifically, TGF-β can form a complex with activin and acts as a constitutive antagonist in cell growth signalling. In addition, serum levels of TGF-β are elevated throughout the entire LR and in vitro induces hepatocyte growth inhibition [70,71] (Figure 1).

The signalling related to the extracellular matrix is another important pathway in this phase. A murine model providing hepatocytes lacking for the integrin-linked kinase (ILK) shows an incomplete termination of the LR. The liver of the ILK KO murine model shows a higher mass than the wild type mouse [72].

In this context, there should also be a remark about apoptosis, which is generally crucial in physiological regulation, and has relevance in the liver. It has already been noted that in a few minutes after PH there is an increase in IL-6 and HGF levels, which protect the hepatocytes from programmed cell death in the first few days. Furthermore, in vivo, apoptosis does not activate during at least the first three days after PH. The activation of caspase 3, which reports the apoptotic reaction, actually occurs after 72 h, simultaneously with the reduction in proliferation. Apoptosis would serve as a brake to the LR, thus allowing the correct attainment of complete liver mass (Figure 1).

Beyond the pathways mentioned, other accessory molecules are known to play a well-defined role into the regenerative process. First requiring further consideration is cyclic adenosine monophosphate (cAMP) [73]. The cAMP signalling pathway controls several genes involved in the process of gluconeogenesis, including: tyrosine aminotransferase (TAT), cAMP-dependent kinase (PKA), CRE-binding protein (CREB), CRE modulator (CREM), CBP/p300, phosphoenolpyruvate carboxykinase (PEPCK), glucose-6-phosphatase (G6Pase) [74,75,76,77]. Since one of the multiple functions of the liver is the distribution of glucose to the whole body, the control of the gluconeogenesis during regenerative phases is extremely important. Specifically, low concentrations of glucagon stimulate DNA synthesis, whereas the opposite effect occurs when concentrations are too high. The levels of cAMP increase considerably after PH, but its role in proliferation is still controversial. Indeed, depending on the concentration of cAMP, there is stimulation or suppression of hepatocyte proliferation [78]. Signalling of cAMP is active throughout the LR process, but its control in proliferation occurs at two points: stimulation of the G0/G1 transition and inhibition near the G1/S phase [79,80]. Hepatocyte odd protein shutting (HOPS), controlled by cAMP, is involved in the early phases of LR. HOPS is a nucleocytoplasmic protein that contributes to the control of cell proliferation by regulating protein synthesis [81]. cAMP regulates the nuclear export of HOPS in normal and regenerating hepatocytes. In cAMP-treated mice, HOPS was detected in the nucleus at 15 min after treatment, while at 60 min, it was fully exported into the cytoplasm. At 90 and 120 min after cAMP treatment HOPS progressively moved back to the nucleus, showing a distribution pattern similar to normal hepatocytes [82]. An interesting and unusual control on post-PH hepatocyte proliferation is played by the glycoprotein CD44, known to bind the hyaluronic acid. The CD44 isoforms containing the exon v6 are modulated during LR, making it a candidate as a possible regulator of hepatocyte proliferation [83].

Another attractive cell source for implementing liver regeneration lies in haematopoietic stem cells. It has been known for several years that these stem cells are able to improve liver parameters following damage. However, it is still not entirely clear how these cells can significantly contribute to the regenerative process. Certainly, from a therapeutic point of view, haematopoietic stem cells are showing significant progress in several clinical trials, with efficacy varying depending on the amount of liver that has been compromised [84,85,86].

## 5. Molecular Mechanisms of Liver Regeneration

Liver regeneration is the product of tight networking between multiple pathways, but surely the two processes that must be modulated involve cell proliferation and cell growth. They can be defined as two sides of the same coin. Specifically, the IL-6/JAK/STAT-3 pathway is involved in proliferation, while PI3K/PDK1/AKT is responsible for cell growth (Figure 1).

### 5.1. Hepatocytes Proliferation

The IL-6/JAK/STAT3 signalling pathway has been well characterised. Specifically, IL-6 produced by the Kupffer cells binds its IL-6 receptor α (IL-6R α), which forms a hetero-hexameric complex with IL-6 receptor-β (gp130). At this step, JAK proteins bind the Box1-2 domains on intracellular gp130. This event mediates the phosphorylation of various tyrosine residues, including those at the C-terminal that serve as binding site for STAT3. STAT3 binds to gp130 and undergoes phosphorylation by JAK to the tyrosine in position 705 (Tyr705), allowing the dimerization of STAT3. Dimeric STAT3 is able to translocate to the nucleus and, after DNA-binding activates the transcription of the target genes, triggers LR [58,87]. Post-translational modifications are also responsible on the negative regulation of STAT3. Specifically, the phosphorylation of serine-727 promotes the de-phosphorylation of Tyr705, de facto inactivating STAT3 [88]. Experiments performed in IL-6^−/−^ and TNF-α^−/−^ mice indicate that both those cytokines are significant triggers of LR. The transgenic mice displayed an erroneous proliferative response already in the early stages. IL-6^−/−^ mice show reduced DNA synthesis, which correlates with a missed STAT3 activation and results in reduced expression of the target genes: AP-1, Myc and cyclin D1 [89,90]. To confirm, the STAT3 KO murine model shows a similar phenotype with a marked reduction in hepatocyte proliferation [91]. Furthermore, when STAT3^−/−^ mice are injected with IL-6 before PH, a normal proliferation and the restoration of function is achieved, highlighting the importance of this cytokine [89]. Considering these results, it is clear that IL-6 plays a primary role in the LR, being directly involved in the activation of STAT3 through gp130 and thus mediates proliferation. Similarly, the KO mice for the TNF-α receptor show an improper activation of NF-κB after PH, which does not allow the production of IL-6 [92]. The KO murine models provide incontrovertible proof of the relevance of these molecules for a successful liver regeneration. In addition to the IL-6/JAK/STAT3 pathway, there are several effectors that enhance mitogenic signalling in hepatocyte proliferation, such as guanylyl cyclase domain containing 1 (GUCD1), which is strongly up-regulated in the first hours after PH [93,94]. The reduction in GUCD1 levels is guaranteed by NEDD4-mediated neddylation, that promotes proteasomal degradation [82]. As well as the bile acids, in appropriate concentrations, also promote cell proliferation, via the farnesoid X receptor (FXR) [95]; or even the Insulin-like Growth Factor 1 (IGF-1), up-regulating HGF and down-regulating TGF-β, contributes to enhanced mitogenic action [96]. These molecules participate in this process, highlighting redundancy as crucial to ensure proper liver regeneration.

### 5.2. Hepatocytes Growth

The regenerative process is given by the integration of several signals and different pathways. Neither strictly essential molecule has yet been identified, and even if the aforementioned KO models lead to an inaccurate result, LR is still initiated. Hepatocyte growth is a complementary mechanism to the proliferation. This arises in the initial phases after PH, when hepatocytes increase in volume and are bi-nucleated [43]. In STAT3^−/−^ mice, after PH, the hepatocytes volume was significantly increased compared to the wild type mice. In addition, a marked phosphorylation of AKT was also observed [58]. Many studies have amply demonstrated that the PI3K/PDK1/AKT pathway is mainly leading to cell growth [97,98]. Phosphatidylinositol 3 kinase (PI3K) is a lipidic kinase that promotes phosphorylation on 3-OH position of phosphatidylinositol 4,5-bis phosphate (PIP3) [99]. The signalling cascade begins with stimulation of the tyrosine kinase receptors (RTKs) which activate in PI3K [100]. Not surprisingly, the most common ligands of RTKs are: HGF, IL-6, TNF-α, TGF-α and EGF [101,102,103]. The phosphatidylinositol-(3,4,5)-trisphosphate (PIP3) production activates 3-phosphoinosistide-dependent kinase 1 (PDK1), which in turn phosphorylates and activates AKT [104]. A negative regulator of the process is the phosphatase and tensin homolog (PTEN), which leads to dephosphorylation of PIP3, inhibiting the signalling cascade [105]. This pathway controls several physiological processes, because PDK1 is responsible for the activation of Protein kinase C ζ (PKCζ), inducing mitogenesis, cellular survival and protein synthesis [106]. AKT activation is obtained by phosphorylation of threonine-308 in the kinase-region and serine-473 in the hydrophobic domain. Both phosphorylation events provide full activation and the correct functioning of AKT in liver regeneration [107,108]. Certainly, the mTOR1 complex (mTORC1) is the main target of AKT, which regulates protein synthesis, depending on nutrient availability and in response to growth factors [109]. Active mTORC1 promotes protein translation by phosphorylating 4E binding protein 1 (4EBP1), preventing binding with eukaryotic translation initiation factor 4E (eIF4E). The unbound factor initiates the translation of proteins involved in growth and cell division, such as cyclin D and c-myc [110,111]. The relevance of AKT signalling has been confirmed in experiments with PDK1^−/−^ mice, in which AKT phosphorylation is reduced and the regenerative process is incomplete [112].

Hepatocytes are clearly the main players in proliferative and growth pathways. Sometimes when the availability of hepatocytes is reduced or otherwise insufficient to support the damage response, alternative pathways exist [113]. In this context, the second cellular component of the liver parenchyma, the cholangiocytes, takes over. Indeed, several models have confirmed that, in the case of impaired hepatocyte regeneration, a response is established that significantly increases the percentage of hepatocytes derived from cholangiocytes, which enables the regenerative process to be correctly completed [114]. The intrinsic plasticity that hepatocytes and cholangiocytes display facilitates the trans-differentiation process, so that one cell type can make up for the lack of the other and vice versa [115,116,117].

### 5.3. IL-17: A Balancer in Liver Regeneration

In recent years, novel molecules involved in LR have been identified, which provide a contribution to the final resolution. Among these, interleukin-17 (IL-17) is certainly the most prominent. The IL-17A, hereinafter IL-17, is the most characterised member of the IL-17 family. This cytokine binds to the receptor for IL-17A (IL-17RA), which dimerizes with IL-17RC, triggering the IL-17 signalling [118]. The activation of this pathway recruits the protein adaptor ACT1, which leads to the activation of relevant pathways such as NF-κB and mitogen-activated protein kinases (MAPK) [119,120]. The cells identified as the first producers of IL-17 are CD4+ Th17. Specifically, Th17 differentiate from naive T cells when influenced by TGF-β, IL-6 and IL-21, which amplify Th17 and induce the expression of the receptor for IL-23 (IL-23R). At this step, IL-23 binds to its receptor and allows the stabilization of the phenotype Th17, which will produce IL-17 [121]. The function of Th17 is offset by regulatory CD4+ T cells (Treg). The presence or absence of inflammation may or may not favour differentiation towards one cell type or the other. Moreover, immune homeostasis also depends on the balance of Th17/Treg and thus on the regulation of the corresponding pathways [122,123,124]. However, Th17s are not the only cells that produce IL-17. In response to different stimuli, other immune cells can produce IL-17, such as γδ T cells, CD8+ T cells and natural killer (NK) cells [125,126]. The pro-inflammatory role of IL-17 is demonstrated in both acute and chronic inflammation. In addition, the activity of IL-17 often requires synergy with other immune molecules. In particular, cooperating with TNF-α, IL-17 induces a massive production of IL-6 and IL-8 in the hepatocytes, which leads to the activation of pro-inflammatory genes [127,128]. The IL-6-promoted inflammatory environment includes neutrophils recruitment and activation of fibroblasts followed by hepatic fibrosis [129,130,131]. IL-17 acts on liver cell types other than hepatocytes. Kupffer cells express IL-17R when stimulated by IL-17 and increase their immunological activity and produce other pro-inflammatory cytokines [132].

In the last decade, several research groups have investigated the role of IL-17 in liver. By using IL-17 KO murine model it has been possible to identify the involvement of cytokine in liver regeneration after PH. IL-17^−/−^ mice, although showing a final recovery similar to their wild type mice, had a reduced proliferative activation at 72 h and 5 days after PH. In particular, using specific proliferative markers such as Ki-67 and BrdU, IL-17^−/−^ hepatocytes exhibit a proliferative latency compared to the wild type hepatocytes. In addition, cyclin D1, a key molecule in the progression from G1 to S phase, was reduced in IL-17 KO respect to wild type mice, confirming an impairing of cell cycle progression in IL-17 deficiency [133]. As mentioned above, γδ T cells are one of the producers of IL-17. During the first hours after PH, the γδ T cells increase in number and return to baseline after about 6 h, suggesting an involvement in the LR. Specifically, TCRδ-KO mice show a marked reduction in regenerative process, demonstrating that γδ T cells play a critical role in LR [134]. The role of IL-17 in LR has also been studied in IL-17R KO murine model. Serum analysis of these transgenic mice showed a reduced concentration of IL-6 compared to controls. IL-6 is crucial in activating STAT3, which controls the gene expression of early genes involved in the priming of residual hepatocytes, whereby in IL-17R^−/−^ mice the regeneration is delayed. The concentration of cyclin D1 in the IL-17R KO models is reduced, explaining the proliferative delay of the hepatocytes. The altered priming of the hepatocytes after PH is related to the slow G1/S transition in IL-17R KO murine model [135]. Interestingly, regulatory genes such as C/EBPβ and A20 are under the control of IL-17R signalling [136]. The gene expression profile, especially in A20, shows a significant reduction in the KO murine model compared to wild type mice, that reaches the peak at 48 h after PH [135]. A20 is a novel identified molecule, which is induced during liver regeneration [137]. A20 regulates IL-6/STAT3 signalling, reducing inflammation and promoting cell proliferation. Its function is performed by down-regulating suppressor of cytokine signalling 3 (SOCS3), which stimulates the activation of STAT3 and thus promotes mitogenesis. In addition, A20 inhibits NF-κB signalling, reducing the activation of pro-inflammatory genes and promoting regeneration [138] (Figure 2).

An interesting feature of IL-17 is that it may also be involved in the termination phase of the liver mass restoring. Considering the very great number of antigens reaching the liver, it is simple to expect high levels of Natural Killer T (NKT) cells in situ [139]. When active, NKTs promote the production of interferon-gamma (IFN-γ), which induces apoptosis in hepatocytes, interrupting liver regeneration [140]. High levels of IL-17 interfere with the NKTs activation, which producing IL-4, stimulates proliferation and allows the liver to reach the ideal mass. At the same time, low levels of IL-17 promote the final phase of LR, stimulating the apoptotic process [134]. Evidences that apoptosis could be involved in ending the proliferation in the restored liver opened up new fields of study. Certainly, involved in the LR, but not well characterised is another cytokine: IL-22. In the regenerating liver, γδT cells are the main producers of IL-22. This cytokine is able to induce in vitro hepatocytes proliferation [141]. In addition, IL-22-mediated signalling includes the JAK/STAT pathway, which provides up-regulation of numerous pro-regenerative genes, such as TNF-α and IL-6, which are essential in LR [142].

The role of IL-17 and immunosuppression is very controversial. Certainly IL-17 is functional in promoting the inflammatory state and autoimmunity, but a protective role in inflammatory dysfunction of the digestive system has also been reported. The role of IL-17 would also appear to be related to the cell type with which it interacts. X. Han et al. report that IL-17 increases immunosuppression in the presence of mesenchymal stem cells, while J. Tian et al. show that IL-17 can downregulate the suppressive capacity of olfactory ecto-mesenchymal stem cells [143,144]. The topic is certainly very fascinating, but requires further study and investigation in order to formulate new hypotheses.

## 6. Recruitment of Immune Cells in Response to Liver Damage: The Balancing Process

The regenerative process occurs not only as a result of loss of liver parenchyma, as after PH, but also as a consequence of exposure to exogenous and/or endogenous agents, such as alcohol and hepatitis B or C viruses, which induce the death of hepatocytes. In the blood or skin there is a constant cell turnover, while in the liver the rate is very low and at homeostasis almost all hepatocytes (approximately 98%) are in a quiescent state. The liver response after injury can be described in consecutive phases, starting with the homeostatic state, followed by inflammation, which could lead regeneration or fibrosis. Ultimately, the regression and resolution phase may occur. Different cell types are recruited during these phases, with a large prevalence of immune cells. At homeostatic stage in the liver, the hepatocytes are in a quiescent state, and there is no activation of any cell type. Liver regeneration is activated when damage is detected. One of the first processes to be activated is inflammation. In fact, damaged hepatocytes are recognised by leukocytes, which help to accentuate the inflammatory state [145]. Kupffer cells play a primary role in the activation of the inflammatory cascade through IL-6 and TNF-α [146,147]. In this phase, the liver objective is to replace the damaged epithelium mainly by increasing the proliferation of hepatocytes, but also by activating the ductal progenitors. Sinusoidal cells support the generation of mitogenic signals such as HGF and Wingless-Type MMTV Integration Site Family Member 2 (WNT2), which allow the expansion of hepatocytes [148]. On the other hand, tissue repair, takes place by means of extracellular matrix (ECM) deposition. The cells responsible for the production of ECM are myofibroblasts and in the liver their precursors, stellate cells. HSCs, activated by macrophages secreting TGF-β and TNF-α, are cells with great replication capacity and express high levels of collagen and tissue inhibitors of metalloproteinases (TIMPs) [149]. As a result of the inflammatory response, bone marrow-recruited monocytes can also differentiate into myofibrocytes and contribute to regeneration [150].

Damage-induced ECM deposition is a transitory phase of the regenerative response, and successful healing requires its subsequent clearance [151]. Thus, a Th1 type immune response is established, which avoids scarring by the production of IL-12 and IFN-γ, opposing the increased levels of TGF-β [152]. Fibrosis is the third level that occurs as a result of excessive accumulation of ECM, which leads to scarring of the liver and loss of function [153]. This phase is extremely critical and must be finely controlled because there is a risk of reaching a tipping point, where the process becomes irreversible and homeostasis of the liver will no longer be guaranteed. Ductal progenitors proliferate exponentially and at the same time the recruitment of monocytes increases, which differentiate into macrophages secreting TGF-β and thrombospondin-1 [154]. A poor balance between TIMPs and matrix metalloproteinases (MMPs) causes the over-accumulation of ECM. Specifically, metalloproteinases degrade the deposited collagen fibres, making fibrosis a reversible process. However, it can happen that activated myofibroblasts produce many TIMPs, which inhibit the action of MMPs, promoting the ECM accumulation [155,156]. Macrophages also require balancing, in particular monocyte-derived macrophages (MoMϕs). Hepatic MoMϕs are divided into two main subsets according to expression levels of Ly6C marker: Ly6C^high^ and Ly6C^low^ monocytes [157,158,159]. Ly6C high MoMϕs represent the dominant population in the early post-damage phases, producing IL-6, IL-1β and TGF-β as pro-inflammatory action is required [160]. To achieve the resolution, after the epithelial damage has been repaired, it is essential to switch the Ly6C high monocytes to Ly6C low. The latter are characterised by an anti-inflammatory phenotype that includes the production of IL-10 and several MMPs, which allows the resolution of fibrosis and the recovery of homeostasis [161,162]. The macrophage switch is determined more by the phagocytosis of apoptotic hepatocytes by STAT3/IL-10/IL-6 signalling pathway [161,163]. In the context of fibrosis an immune response of type Th2 is involved, where inflammatory environment is enhanced by an increase of IL-4 and IL-13 and myofibroblasts deposit an excessive extracellular matrix [164,165].

In the duality of regeneration and fibrosis, LSECs are strongly influenced by the stimuli released by the surrounding microenvironment. In particular, in the regeneration phase they express markers such as CXCR7 and CXCR4, which induce the repopulation of hepatocytes, while in the case of fibrosis they express CXCR4 in great amount, which promotes the secretion of profibrotic cytokines [166]. The last phase of this cycle is the regression of fibrosis and resolution of the damage, which brings the liver to its physiological condition. During regression phase, different cell types are recruited into the tissue, such as dendritic cells (DCs), macrophages and NKs. Their role is to restore the status quo before the fibrotic state development and allow the resolution. DCs through the secretion of MMPs, specifically MMP9, degrade the deposited ECM, preventing the scar formation [167,168]. In this phase, the inactivation of myofibroblasts and their subsequent elimination is essential. This function is carried out by the NKs, who preferably kill the MICA-ligand expressing senescent HSCs, through recognition by NKG2D receptor. In addition, the expression of the NKp46 ligand on myofibroblasts may also trigger cytotoxic activity mediated by NK cells, reducing liver fibrosis [169,170,171]. The role played by γδ T cells remains controversial, on one side the cytotoxicity against HSC induces a protective role, on the other the production of IL-17A gives a harmful effect [172,173] (Figure 3). 

Characterisation of γδ T cells over the next few years will make it possible to gain new insights into their function as a bridge between innate and acquired immunity. Moreover, the action of these would also appear to be strongly influenced by the microbiota [174]. Recently, an axis of interaction between the liver and the gut microbiota has been well defined. In particular, it is known how both support and control various functions for the other. Such as the mechanism of bile production, in which both are involved through feedback mechanisms. However, the regulatory mechanism that the microbiota would have on the liver regeneration process itself is not well known. In chronic diseases, it is known that tissue degeneration leads to or is driven by an increased inflammatory state, in which the gut microbiota may be one of the main modulators. In the event of destruction of the intestinal barrier, microorganisms or their toxins can easily reach the liver, via the portal vein, dramatically affecting its functions. Undoubtedly, the identification of the interaction between the immunological component in the liver-intestine axis will bring new revelations in the coming years that may also favour therapeutic approaches [175,176,177,178,179].

## 7. Conclusions

The study of liver regeneration is clinically relevant to investigate liver transplantation or hepatocarcinoma in human, it also provides a powerful model for studying in vivo the pathways of proliferation and growth. The successful process is guaranteed by the integration of different mechanisms, from the action of specific molecules to well-established timescales. Curiously, the LR is like the *Staircase of San Patrizio* in Umbria (Italy). The two stairways, up and down, do not cross each other, albeit in acoustic communication, but both serve to reach the same place/target. Likewise, during liver mass restoring, cell proliferation and growth are on one side, sustained by pro-inflammatory immunity, while the remodelling process helped by anti-inflammatory measures is on the other; the equilibrium retains and preserves the physiology and liver function, thus guaranteeing the homeostasis of the organism. The early phases of the regeneration process have certainly been better characterised and it is true that “a good beginning is half the battle”, but in this specific case the terminal phase is equally important. Indeed, the ending of regeneration by the hepatostat after basal values have been restored is a critical point. One of the inhibitory signals could be the activation of the apoptosis process, which would stop cell division and growth. Indeed, to the well-known molecules involved, such as IL-6 and TNF-α, IL-17 undoubtedly covered an important and strategic role in LR. This cytokine is present throughout the regenerative cycle. Effectively, it has been seen as important as contributing to proliferative triggering in the early stages. However, IL-17 role in termination phase is still poorly explored. At this stage, the reduction of IL-17 levels would allow NKT cells to produce INF-γ, thus promoting apoptosis. In the future, the characterisation of IL-17 will be essential. This cytokine is definitely a mediator of inflammation, and its synergetic action with other stimuli makes it crucial in those processes where the action of several mediators is required, such as liver regeneration.

A further contribution in recent years has been made by the development of new in vitro devices, which are proving to be a valid alternative for in vivo research and allow faster characterisation of cell interactions in different biological contexts. These include approaches of both 2D and 3D culture devices, which exploit innovative biomaterials with different properties depending on the type of application. For instance, the use of polymer replicas, such as polycarbonate (PC) or polydimethylsiloxane (PDMS), to which extracellular matrix proteins such as collagen or laminin can be implemented, making the device ideal for studying cellular organisation and differentiation or assessing shear stresses or vascularisation [180,181,182,183,184]. Even more recent are organ-on-a-chip devices, which interest not only the liver, but also other organs and tissues such as heart and kidney [185]. Among the biochips in the hepatic field, we find specific devices for the characterisation of pathological conditions such as NAFLD [186,187,188] or ALD [189,190,191], but also sophisticated microfabricated platforms that allow the assessment of cellular spatial location in accordance with different metabolic levels, which are created ad hoc by applying microfluidics [192].

## Figures and Tables

**Figure 1 ijms-24-01176-f001:**
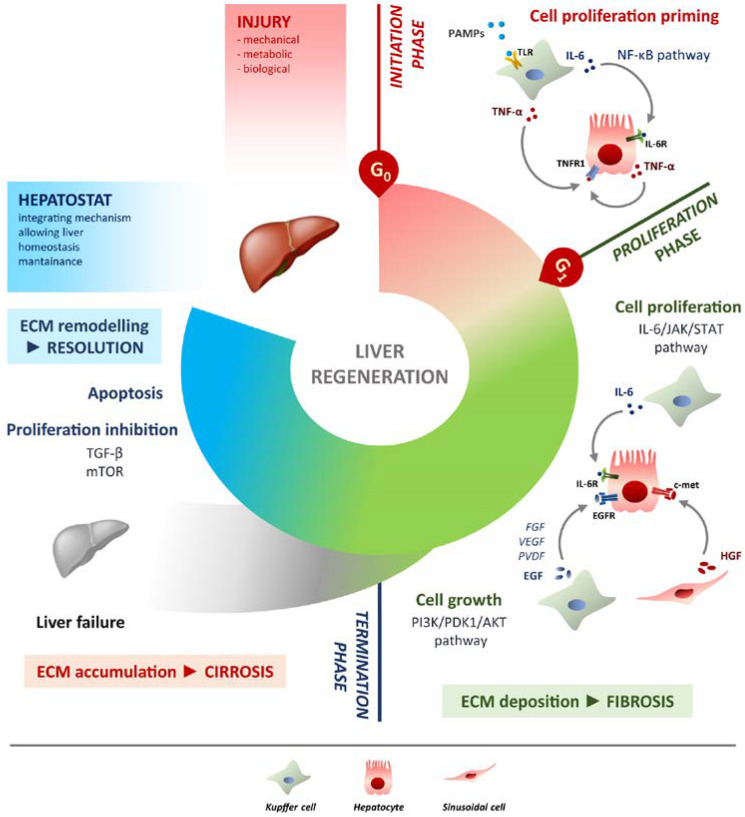
Schematic overview of liver regeneration. LR outlined in three phases: initiation, proliferation and termination. In the initiation phase different PAMPs trigger regenerative process by interacting with TLRs exposed on Kupffer cells membrane. TNF-α and IL-6 produced by Kupffer cells activates the NF-κB pathway in hepatocytes making them responsive to the proliferative signalling and forthcoming to G0/G1 transition. Once IL-6 determined the activation of proliferation, acting on JAK/STAT3 pathway, hepatocytes enter the G1 phase and growth factors signalling ensures their progression in the cell cycle. Kupffer cells and sinusoidal cells sustain hepatocytes growth by producing respectively HGF acting through c-met receptor and EGF. EGFR ligands other than EGF (FGF, VEGF, PDGF and others) act as signal modulators. Growth factors binding to their specific receptors lead to cell growth mainly through the PI3K/PDK1/AKT pathway. The termination phase takes place when the original weight of the liver is reached and results from inhibition of proliferation and growth, and apoptosis induction. The mTOR pathway controls protein translation and together with TGF-β induces hepatocyte expansion inhibition. Apoptosis cooperates with other inhibitory signals, allowing the correct attainment of liver mass. The hepatostat is responsible for the perfect integration of such processes leading liver structural and functional recovery. ECM deposition is a transitory and reversible condition of the regenerative response that arises during the proliferative phase and leads to fibrosis. ECM accumulation requires its rapid and continuous degradation and remodelling to ensure resolution, avoiding cirrhosis outcome and/or liver failure.

**Figure 2 ijms-24-01176-f002:**
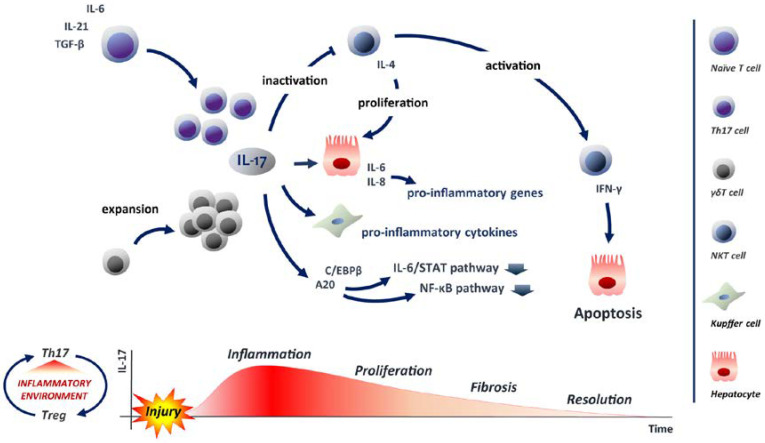
IL-17 impact in liver regeneration IL-17 is mainly produced by Th17 - from naïve T cells differentiation upon TGF-β, IL-6 and IL-21 stimulation - and γδ T cells. Inflammatory environment induced by injury sustains Th17 population and controls the balance between Th17 and their counteracting Treg cells to preserve immune homeostasis. Upon injury γδ T cells increase in number to sustain IL-17 levels. IL-17 acts on hepatocyte inducing massive production of IL-6 and IL-8 to trigger pro-inflammatory genes activation. IL-17 induced Kupffer cells produce pro-inflammatory cytokines. IL-17 signalling controls C/EBPβ and A20 regulating in turn IL-6/STAT3 and NF-κB signalling. In proliferating environment high level of IL-17 interferes with NKs activation inducing IL-4-mediated proliferation; reduced IL-17 levels at termination phase promote IFN-γ production by NKs reactivation, inducing apoptosis in hepatocytes and interrupting regenerating process.

**Figure 3 ijms-24-01176-f003:**
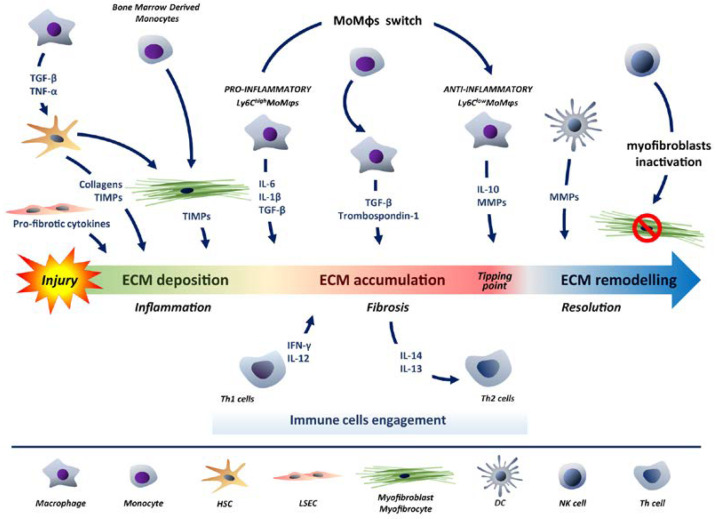
Immune cells engagement in liver regeneration. Hepatic damage induces regeneration, fibrosis and finally implies regression to achieve resolution. Tissue repair takes place by means of ECM deposition. HSCs derived myofibroblasts are primarily responsible for the production of ECM through TIMPs secretion. Secondly, inflammatory background induces bone marrow-recruited monocytes to differentiate into myofibrocytes. LSECs contributes to repair producing pro-fibrotic cytokines. Excessive deposition of ECM induces fibrosis that, if out of control, leads to a tipping point making irreversible ECM accumulation and resulting in liver failure. Monocyte-derived macrophages contribute to ECM accumulation by secreting TGF-β and thrombospondin-1. The two main subsets of hepatic macrophages (MoMϕs) are classified according to Ly6C expression. Ly6C^high^ MoMϕs produce pro-inflammatory IL-6, IL-1β and TGF-β in the early phases, to switch to Ly6C^low^ anti-inflammatory phenotype producing IL-10 and several MMPs, allowing for ECM remodelling and the recovery of homeostasis. DCs through the secretion of MMPs, degrade the deposited ECM, preventing the scar formation and making fibrosis a reversible process. NKs carried out inactivation of myofibroblasts and their subsequent elimination. Inflammatory background engages Th1 and Th2 immune response to balance ECM deposition and remodelling.

## Data Availability

Not applicable as this review did not report new data.

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
