# Peer review of "Liver Regeneration and Immunity: A Tale to Tell"

_ijms, 2023, doi:10.3390/ijms24021176_

Round 1

Reviewer 1 Report

The authors reviewed the relationship between liver regeneration and immunity by identifying the molecular mechanisms involved in the interplay between proliferation and inflammation.

Despite the fact that the paper is well-written and the topic is interesting, the literature they examined is too old to provide the readers with the most recent advances in that field. Furthermore, several reviews on that topic have recently been published in the last years that discuss the same topics as this review. 

(Li N, Hua J. Immune cells in liver regeneration. Oncotarget. 2017 Jan 10;8(2):3628-3639.; Li H, Zhang L. Liver regeneration microenvironment of hepatocellular carcinoma for prevention and therapy. Oncotarget. 2017 Jan 3;8(1):1805-1813.; Qian Y, Shang Z, Gao Y, Wu H, Kong X. Liver Regeneration in Chronic Liver Injuries: Basic and Clinical Applications Focusing on Macrophages and Natural Killer Cells. Cell Mol Gastroenterol Hepatol. 2022;14(5):971-981.; Tang C, Chen H, Jiang L, Liu L. Liver Regeneration: Changes in Oxidative Stress, Immune System, Cytokines, and Epigenetic Modifications Associated with Aging. Oxid Med Cell Longev. 2022 Jul 28;2022:9018811.; Hu C, Wu Z, Li L. Mesenchymal stromal cells promote liver regeneration through regulation of immune cells. Int J Biol Sci. 2020 Jan 22;16(5):893-903).

Also, the gut microbiota's role deserves to be discussed better.

Author Response

Dear Editor,

please find enclosed the rebut to the reviewers. We thank the reviewers that with their comments improved our manuscript.

Reviewer 1

  1. The authors reviewed the relationship between liver regeneration and immunity by identifying the molecular mechanisms involved in the interplay between proliferation and inflammation.

Despite the fact that the paper is well-written and the topic is interesting, the literature they examined is too old to provide the readers with the most recent advances in that field. Furthermore, several reviews on that topic have recently been published in the last years that discuss the same topics as this review. 

(Li N, Hua J. Immune cells in liver regeneration. Oncotarget. 2017 Jan 10;8(2):3628-3639.; Li H, Zhang L. Liver regeneration microenvironment of hepatocellular carcinoma for prevention and therapy. Oncotarget. 2017 Jan 3;8(1):1805-1813.; Qian Y, Shang Z, Gao Y, Wu H, Kong X. Liver Regeneration in Chronic Liver Injuries: Basic and Clinical Applications Focusing on Macrophages and Natural Killer Cells. Cell Mol Gastroenterol Hepatol. 2022;14(5):971-981.; Tang C, Chen H, Jiang L, Liu L. Liver Regeneration: Changes in Oxidative Stress, Immune System, Cytokines, and Epigenetic Modifications Associated with Aging. Oxid Med Cell Longev. 2022 Jul 28;2022: 9018811.; Hu C, Wu Z, Li L. Mesenchymal stromal cells promote liver regeneration through regulation of immune cells. Int J Biol Sci. 2020 Jan 22;16(5):893-903).

We follow the comment of the reviewer 1 and we updated the references.

  1. Also, the gut microbiota's role deserves to be discussed better.

As rightly suggested from the reviewer 1, we improved liver-microbiota axis.

We think to have adequately reply to the request of the reviewers and we again thank them to render the manuscript more interesting and readable.

Reviewer 2 Report

1. As the liver is the main target organ, the authors discuss the roles of Kupffer cells and hepatocytes in liver regeneration and immunity. However, the roles of stellate and bile duct cells in liver regeneration and immunity were not mentioned. 

2. Discuss the role of Hematopoietic stem cells and liver regeneration.

3. Recently, new in vitro models, such as microfabricated cell culture devices, have been widely applied for research. Please discuss them in the review and their usage.

4. Line 32, additional and new references are needed.

5. Reference needed in Figure 1, 2 and 3 legends.

6.      Flow of the review article could be better (minor)

Author Response

Dear Editor,

please find enclosed the rebut to the reviewers. We thank the reviewers that with their comments improved our manuscript.

Reviewer 2

Comments and Suggestions for Authors

  1. As the liver is the main target organ, the authors discuss the roles of Kupffer cells and hepatocytes in liver regeneration and immunity. However, the roles of stellate and bile duct cells in liver regeneration and immunity were not mentioned.Hepatic stellate cells (HSC) are mentioned in the introduction and in paragraph 6, about the recruitment of immune cells in response to liver damage. Moreover, they are in Figure 3 in the ECM deposition contest.

On the base of the comments of the reviewer 2, we updated the involvement of cholangiocytes functionality in the line 319-328.

Discuss the role of Hematopoietic stem cells and liver regeneration.

Also in this case, we added in line 248-254 of the manuscript the request of discussion on the base of the comment of reviewer 2.

  1. Recently, new in vitro models, such as microfabricated cell culture devices, have been widely applied for research. Please discuss them in the review and their usage.

We discuss, as requested, the new in vitro models of cell culture device, we described them from line 512 to 525 of the manuscript.

  1. Line 32, additional and new references are needed.

In line 32 references are added [1-5]

  1. Reference needed in Figure 1, 2 and 3 legends.

The figures are a suggested model of cells interaction according with the references in the text.

We think to have adequately reply to the request of the reviewers and we again thank them to render the manuscript more interesting and readable.

Reviewer 3 Report

This is a review on immune factors and liver regeneration, focusing on the signal pathways of hepatocyte proliferation and growth, as well as the immune factors involved in the regeneration process. It is rich in content and clear in order. However, I have a few questions.

1. In the acute and chronic injury of liver, the author mainly introduced the factors of liver damage such as drugs, hepatitis virus, obesity, etc. What about the regeneration mechanism of traumatic liver injury? I think this is an important factor in liver transplantation.

2. The review introduces the evidence of liver regeneration after liver transplantation in animal experiments, line197-200. Are these experimental animals treated with anti-rejection drugs after liver transplantation? If so, what is the effect of anti-rejection drugs on immune factors? Especially the IL-6 and IL-17 pathways highlighted in this article? What about the data of human liver transplantation?

Author Response

Dear Editor,

please find enclosed the rebut to the reviewers. We thank the reviewers that with their comments improved our manuscript.

Reviewer 3

This is a review on immune factors and liver regeneration, focusing on the signal pathways of hepatocyte proliferation and growth, as well as the immune factors involved in the regeneration process. It is rich in content and clear in order. However, I have a few questions.

  1. In the acute and chronic injury of liver, the author mainly introduced the factors of liver damage such as drugs, hepatitis virus, obesity, etc. What about the regeneration mechanism of traumatic liver injury? I think this is an important factor in liver transplantation.

The request of the reviewer 3 is plausible, but we have not introduced the liver transplantation to do not enlarge the panorama of the review.

  1. The review introduces the evidence of liver regeneration after liver transplantation in animal experiments, line 197-200. Are these experimental animals treated with anti-rejection drugs after liver transplantation? If so, what is the effect of anti-rejection drugs on immune factors? Especially the IL-6 and IL-17 pathways highlighted in this article? What about the data of human liver transplantation?

Are these experimental animals treated with anti-rejection drugs after liver transplantation?

Yes, with a short course of antimetabolite therapy, such as cyclophosphamide, which targeted the B-cell proliferative response. Moreover, an immunosuppressive treatment was provided.

If so, what is the effect of anti-rejection drugs on immune factors?

The patient showed a strong immunosuppression after the treatment, which was effective in preventing the rejection.

We think to have adequately reply to the request of the reviewers and we again thank them to render the manuscript more interesting and readable.

Round 2

Reviewer 1 Report

The paper has been improved sufficiently. 

Author Response

We thank the reviewer that with his comments improved our manuscript.

Reviewer 3 Report

The patient showed a strong immunosuppression after the treatment, which was effective in preventing the rejection. 

The authors introduced the importance of IL-17 in promoting liver regeneration. Does the application of immunosuppressants inhibit or promote the production of IL-17? If the production of IL-17 is inhibited, is it contrary to the conclusion of the review? Please explain this problem.

Author Response

Reviewer 3

Comments and Suggestions for Authors

The patient showed a strong immunosuppression after the treatment, which was effective in preventing the rejection.

The authors introduced the importance of IL-17 in promoting liver regeneration. Does the application of immunosuppressants inhibit or promote the production of IL-17? If the production of IL-17 is inhibited, is it contrary to the conclusion of the review? Please explain this problem.

We added some extra consideration about IL-17 and immunosuppression (Lane 415-422).

A further important aspect to consider is the specific presence of receptors for IL-17, which are clearly responsible for triggering and thus the final response. Nevertheless, in this review an attempt has been made to approach the role of IL-17 in liver regeneration by interconnecting known pathways. For example, the section 'IL-17: a balancer in liver regeneration' aims to highlight how IL-17 levels, depending on whether they are high or low, are able to drive proliferation, accentuating the inflammatory state, or induce apoptosis, respectively.